# Arbuscular Mycorrhizal Fungi Associated with Maize (*Zea mays* L.) in the Formation and Stability of Aggregates in Two Types of Soil

**DOI:** 10.3390/microorganisms11112615

**Published:** 2023-10-24

**Authors:** Juan Florencio Gómez-Leyva, Miguel Angel Segura-Castruita, Laura Verónica Hernández-Cuevas, Mayra Íñiguez-Rivas

**Affiliations:** 1División de Estudios de Posgrado e Investigación, Tecnológico Nacional de México/Instituto Tecnológico de Tlajomulco, Maestría en Ciencias en Agrobiotecnología, Tlajomulco de Zúñiga 45640, Mexico; jfgleyva@hotmail.com (J.F.G.-L.); fungicuevas@hotmail.com (L.V.H.-C.); ma_y55@hotmail.com (M.Í.-R.); 2Maestría en Ciencias en Agrobiotecnología, Tlajomulco de Zúñiga 45640, México

**Keywords:** *Claroideoglomus claroideum*, *Rhizophagus aggregatus*, soil classification

## Abstract

Knowledge of native Arbuscular Mycorrhizal Fungi (AMF) and their relationship with the edaphic characteristics where they live is important to establish the influence of allochthonous AMF, which were inoculated, on the development and stability of soil aggregates. The objectives of this research were to know the composition of native AMF species from two contrasting soils, and to establish the development and stability of aggregates in those soils with corn plants after inoculating them with allochthonous AMF. The experiment had three factors: Soil (two levels [S1 and S2]), HMA (three levels: without application [A0], with the application of *Claroideoglomus claroideum* [A1] and with the application of a consortium [A2]) and Fertilization (two levels (without fertilization [f0] and with fertilization [f1])). Twelve treatments were generated, with five replicates (60 experimental units [EU]). The EU consisted of a pot with a corn plant and the distribution was completely random. The results demonstrated that the *Typic Ustifluvent* presented nine species of native AMF, while the *Typic Dystrustert* had three; the native AMF in each soil influenced the activity of allochthonous AMF, such as their colonization and sporulation. Likewise, differences were found in the stability of macro-sized aggregates (0.5 to 2.0 mm).

## 1. Introduction

Arbuscular mycorrhizal fungi (AMF), one of the most abundant microbial groups in soils, are capable of creating mutualistic associations with plants to improve water and nutrient utilization [1], reduce stress caused by high temperatures, regulate biotic and abiotic stress [2,3], improve the physical [4] and microbiological [5] conditions of soils, reduce the absorption of heavy metals by plants [6] and even increase organic carbon reserves in soils [7]. The physical and chemical characteristics of soils are determinants in the composition of AMF populations: soil texture is related to the differences between AMF communities [8]; pH influences the development of AMF populations at a local scale [9]; and the phosphorus (P) contents regulate their presence—at lower contents, the AMF population increases while at higher contents it decreases [10]. Likewise, it has been shown that high electrical conductivity (EC) and high contents of Ca^2+^, Mg^2+^, K^+^ and Na^+^ decrease the sporulation of mycorrhizal fungi [11]. On the other hand, it has been proved that different soil types harbor distinct AMF communities [12], although it has also been suggested that the species composition of AMF communities is related to the plant host composition of different ecosystems [13]. The relationship between plant and AMF communities could differ due to changes in the biotic and abiotic environments [14]. In this regard, Schappe et al. [15] stated that both plant and AMF communities are affected, directly or indirectly, by environmental conditions. This means that the type of climate, the type of soil, the type of microorganism and even the type of plant at a site would influence the plant–AMF association. In the case of climate, changes in temperature and precipitation occurring along an altitudinal gradient [16], as well as solar radiation [17], either favor or limit the functions of the plant–AMF association. Additionally, the physical and chemical properties of the soil influence the spore abundance and AMF colonization [18]. The presence of saprotrophic microorganisms can significantly affect AMF activity [19] and vice versa [20]. However, it has also been reported that the mixture of AMF with other microorganisms (bacteria or fungi) favors the absorption of nutrients by plants [21,22], and the control of phytopathogenic fungi [23,24]. AMF has been studied under different scenarios, for example, under controlled conditions [25,26], in greenhouses [27], in experimental fields [4,28,29], regions [12,16] and even on a global scale [30]. However, Bueno et al. [16] indicated that the study of AMF–plant associations in small areas could be useful to have a better understanding of their interactions by providing detailed information about the suitability of this symbiosis with specific hosts and edaphoclimatic conditions. In this regard, one of the plants that greatly benefits from symbiosis with AMF is maize (*Zea mays* L.), to such a degree that it is considered a volunteer plant [31]. Several studies have been conducted regarding the maize–AMF association, showing that this species is highly mycotrophic [32]. It has been established that, when a soil sown with maize is inoculated with AMF, and the symbiotic relationship is established, the stability of the aggregates increases [4,33]; the stability increases in the surface layer of the soil but decreases as the depth increases [18]. For their part, Oehl et al. [12] indicated that native or endemic AMF communities influence the physical and chemical properties of the soils where they grow to such an extent that soils could be characterized according to the AMF they contain, though this has not yet been demonstrated. The above would mean that the native AMF of a specific soil would influence the activity of the allochthonous AMF strains with which it was intended to be inoculated, modifying the properties of this soil (like the stability of the aggregates). However, information on this is scarce. In this sense, considering that each individual soil is characterized by unique properties closely related to the environmental conditions of a site [34], it is likely that soils with contrasting characteristics and properties, planted with the same type of crop (such as maize) in the same region, would present different communities of native AMF. Therefore, these soils would respond differently to inoculation with AMF from allochthonous strains in such a way that the development and stability of the aggregates would be different between the two soils. Thus, the objectives of this research were to know the species composition of native AMF communities in two contrasting soils of a region and to establish the development and stability of the aggregates in those soils contained in pots as substrates for maize plants in greenhouses, after inoculation with allochthonous AMF.

## 2. Materials and Methods

### 2.1. Research Area

The work was done in the municipality of Tlajomulco de Zúñiga, Jalisco (located between the coordinates 20°21′ N, 103°11′ W and 20°37′ N, 103°38′ W) with soils from an agricultural area. The climate, according to Garcia [35], is semi-warm with summer rainfall [(A) C(wo)], with an average annual temperature of 19.3 °C and a total annual precipitation of 782.7 mm [36]. The humidity and temperature regimes are Ustic and Thermal, respectively (Figure 1).

The rocks at the site were mainly tuffs, basalts, andesites and obsidian, as well as pyroclastic deposits [37]. In the uncultivated areas, mesquite and scrubland could be found. The IIEG-Jalisco [38] defined the existence of four soil groups: Cambisols, Feozems, Luvisols and Vertisols. The main crop in this area is maize, but agave and forced production of red berries could also be found [38].

Soils survey: The selection of sites for soil sampling was done considering soils with contrasting textural classes (clayey and sandy), based on the edaphological map of the municipality [38]. The first site (S1) was placed in the paddock “el algodón”, which is located at 20°26′47″ N latitude and 103°29′18″ W longitude, at 1508 m.a.s.l., and the second (S2) was placed in the locality “Lomas de Tejada” at an altitude of 1568 m, which is located between 20°28′01″ N latitude and 103° 24′27″ W longitude.

### 2.2. Methodology

The methodological phase consisted of seven stages, which are described below:

#### 2.2.1. Characterization and Classification of Experimental Soils

The soil at each site was characterized. First, in the field, the site and soil profile were described according to the USDA Soil Survey Manual [39]. After the description, soil samples (2 kg) from each of the horizons were collected and taken to the laboratory. Additionally, at each site, soil (approximately 600 kg) was collected from the surface layer (0–30 cm depth) to be used as substrate for the maize potting stage. Afterward, the soil of the horizons was dried at room temperature and it was sieved with a 2.0 mm diameter mesh. The physical and chemical properties of the soils were determined for classification purposes, in accordance with the Official Mexican Standard NOM-021-RECNAT-2000 [40], among which are color (dry and wet), mechanical particle analysis (pipette method), bulk density [Bd] (paraffin cloud method), organic matter content (wet combustion), pH (in water 2:1), electrical conductivity (EC) (conductivity meter), carbonate content (by acid neutralization), cation exchange capacity (CEC), exchangeable [ammonium acetate pH 7, 1N] and soluble (from saturated paste extract) cations and phosphorus pentoxide (P_2_O_5_) content [citric acid method]. Total nitrogen (Ns) was determined in the surface layer by the Kjeldahl method and available phosphorus (Ps) by the Bray and Kurtz procedure 1. Once the laboratory data and field information were obtained, soil classification was made according to the Keys to Soil Taxonomy [41]. Separately, topsoil samples (100 g) were used to extract and quantify the spores of native AMF in the study soils, using the wet sieving method [42]. The obtained spores were mounted on slides with polyvinyl alcohol–lactic acid glycerol (PVLG) or PVLG mixed with Melzer’s reagent in a 1:1 (*v*/*v*) ratio and left to dry for 24 h at room temperature for microscopic observation; the morphological characteristics of the spores were compared and contrasted with specialized descriptions of Glomeromycota [42,43,44,45] to determine their taxonomic identity. Species nomenclature was defined based on the work of Redecker et al. [46]. The soils to be used as substrate were dried under shade and at room temperature and, when dry, they were taken to a greenhouse to be used for filling pots.

#### 2.2.2. Biological Material

The vegetative material used for mycorrhization with AMF (native and allochthonous) was a maize hybrid (H-SCS15) of intermediate and rainfed cycle. This material was chosen because it is grown in the region. The seeds had not been treated with antifungal products. The allochthonous AMF with which the soil was inoculated were *Claroideoglomus claroideum* and a consortium composed of six AMF species (*Acaulospora excavata*, *Acaulospora kentinensis*, *Acaulospora morowiae*, *Acaulospora scrobiculata*, *Funneliformis mosseae* and *Sclerocystis sinuosa*).

#### 2.2.3. Fertilization

The fertilizer was applied following the fertilization recommendation for maize in the research area [47]. The fertilizer was added in 2 stages, the first at planting, where the amount corresponding to the dose of 300 kg ha^−1^ of diammonium phosphate (DAP) was applied, and the second 40 days after planting, where 300 kg ha^−1^ of urea was added. This was done to evaluate the activity of AMF in soils with and without fertilizer.

#### 2.2.4. Experimental Design

The experiment consisted of three factors: Soil (S), AMF (A) and Fertilization (f), where S had two levels (soil site 1 [S1] and soil site 2 [S2]), A had three levels (without application of AMF [A0], with application of *Claroideoglomus claroideum* [A1] and with application of the consortium [A2]) and f had two levels (without fertilization [f0] and with fertilization [f1]), being 12 treatments in total (S1A0f1, S1A1f1, S1A2f1, S1A0f0, S1A1f0, S1A2f0, S2A0f1, S2A1f1, S2A2f1, S2A2f1, S2A0f0, S2A1f0 and S2A2f0). Each treatment had five replicates for a total of 60 experimental units (EU). Each EU consisted of a pot with a maize plant under the corresponding treatment. The distribution of the EU was completely randomized.

#### 2.2.5. Experiment Setup

The experiment was conducted under protected conditions (curved roof greenhouse). The pots used to make the EUs had a 20 L capacity (height 19.5 in, diameter 9 in). To carry out the sowing, the pots were first filled with S1 soil, previously homogenized in each of the corresponding treatments and replicates. The same procedure was followed for S2 soil. Once the soil had been placed in each pot, water was added to bring them to saturation. The pots with soil and water were left to drain in the greenhouse until they reached field capacity (which was detected with an Extech moisture meter), in order to start planting. Maize seeds (three) were sown in each pot, after which the soil of each pot was inoculated with 50 g of inoculum, with allochthonous AMF corresponding to 800 spores per gram of soil. Fertilizer in the fertilized treatments was added as indicated above. It is important to clarify that the amount of fertilizer for each soil was different; the calculation was done considering the Bd of each soil. Seven days after the emergence of the maize seedlings in the pots, thinning was carried out to leave only one plant per pot. The plants were irrigated with one liter of water every third day at eleven o’clock in the morning.

#### 2.2.6. Variables Evaluated

The evaluation was done on the potting soil and on the plant. The soil variables were aggregate stability (AS), total nitrogen (Ns), available phosphorus (Ps), AMF colonization in maize secondary roots (Co) and AMF spore density (Sp). The following were determined in plant tissue: total nitrogen in plant (Np), phosphorus in plant tissue (Pp) and yield (Yd).
The AS was determined in the soils before applying the treatments and at the end of the experiment (potting soil); the AS was determined by wet sieving and air-drying method (2 to 0.5 mm) [41].Ps was determined by the Bray and Kurtz method 1, Pp by colorimetry with nitro-vanadomolybdate and Ns in soil and plant were obtained by the Kjeldahl method [38], both at the end of the experiment.Co by AMF was obtained by differential staining technique with trypan blue [48] at the end of the experiment.Sp was obtained in 100 g of soil by direct counting of AMF spores extracted from the study soils using the wet sieving method [42] at the beginning and the end of the experiment.The HP was measured with a flexometer, the measurement was taken from the soil surface to the highest part of the plant while the plant diameter was measured with a vernier at 20 cm from the soil surface before harvesting.

#### 2.2.7. Statistical Analysis

To determine the effect of the treatments on the study variables, the results were subjected to an analysis of variance (*p* = 0.05). In those variables where differences were detected, a comparison of means was carried out (Tukey, *p* ≤ 0.05). In addition, the effect of the factors on the variables evaluated was established with a factorial analysis (*p* ≤ 0.05). A Pearson correlation was established (value *p* = 0.05) as well. Analyses were performed with Minitab 17 software [49].

## 3. Results

### 3.1. Soils under Study

The low organic carbon contents (<6%), the thickness of the horizons (Table 1) and the firm consistency of the surface layer aggregates (epipedon) allowed the identification of the ochric diagnostic horizon in both soils. The soils did not show endopedons. S1 showed vertic properties and, therefore, was classified in the Vertisols order, while S2 was classified in the Entisols order. By considering the soil moisture and temperature regime (Ustic and Thermal, respectively), particle size, mineralogy, CEC and reaction (pH), at the family level, S1 was classified as fine, mixed, semiactive, thermic Typic Dystrustert; S2, on the other hand, was classified as a fine-loamy, mixed, superactive, nonacid, thermic Typic Ustifluvent.

The AMF communities of the soils were composed of species from taxonomic genera different from those of the allochthonous AMF used in the inoculums. In S1, three AMF species were identified (Table 2) which appeared among the nine found in S2 (Figure 2). The native AMF population per gram of soil in S1 and S2 was 520 and 460 spores, respectively. The presence of a higher richness of native fungi in S2 indicates the influence of soil characteristics. When considering the number of species and spores in each soil, as well as the altitudinal position of each site, a relationship between the soils, number of communities, number of spores and altitude was observed, since the altitudinal difference between S1 and S2 is 60 m, with S2 being found at the lowest point (1508 m).

### 3.2. Mycorrhizal Fungi in Soils

Colonization by AMF on maize roots (Figure 3) presented differences (Tukey, *p* ≤ 0.05) between treatments (Table 3). In this sense, treatment S1A1f0 had the highest colonization (97.33%), a result similar to the 96.76% of S1A2f0. Both percentages were higher than those obtained in S1A0f0 (70.66% and 59.32%, respectively). These results are noteworthy because, even though the soil was the same, the effect of the application of the AMF inoculum (*Claroideoglomus claroideum* or consortium) and the fertilizer on the colonization were observed and fertilization caused a decrease in colonization in both soils. However, treatment S2A0f0 presented a colonization of 96.67%, a percentage similar to S1A1f0 and S1A2f0 and hence higher than S1A0f0, reflecting the different behavior of colonization according to the soil and the presence of native AMF. This was evident in the factorial analysis (Table 3), where the soil type, AMF and fertilizer were involved in colonization (*p* = 0.000). However, if the value of the *F* statistic is considered, the fertilizer had a greater influence (*F* = 30.48) than the soil type and inoculum (*F* = 23.38 and *F* = 18.15, respectively); that is, fertilization negatively affected colonization by AMF. In this sense, the *p* value of the interaction S-A-f indicates that this interaction is significant for the colonization behavior, showing that fertilization affects colonization (Figure 4), inoculation with *Claroideoglomus claroideum* and when using a soil substrate such as S2. The relation between the factors and the S-A-f interaction explained 73.51% of the AMF colonization behavior in maize plants.

Regarding spore density, all three study factors influenced the results (*p* = 0.000); however, the factor that had the greatest effect on the number of spores was inoculation with AMF (*F* = 2111.49). According to the main effects analysis, the use of S2, inoculation with *C. claroideum* and fertilizer application (Figure 4b) had a greater effect on sporulation, which caused the treatments to present differences (Tukey, *p* ≤ 0.05) in the number of spores (Table 1), with the highest number of spores present in S2A1f0 (3153.00) and the lowest in S1A0f0 (866.00).

### 3.3. Aggregates Stability

Aggregates stability between treatments (Table 3) showed differences (Tukey, *p* ≤ 0.05). Treatment S1A1f0 presented the highest percentage of stable aggregates (79.76%), whereas S2A0f1 had the least stable aggregates (46.21%). These results again reflect the influence of the soil type, AMF and fertilizer on AS (Table 4), for example, AS had a positive correlation (0.537) with sporulation (Table 5). Each of these factors affected the AS, with soil having the greatest effect (*F* = 573.96). This was confirmed by the graph of the main effects on AS (Figure 3) since soil type S1 mainly affected AS, followed by *C. claroideum* and no fertilization, for AMF and fertilizer factors, respectively. The relation between the soil types, AMF and fertilizer, as well as the S-A interaction, explained 94% of the aggregates stability behavior.

### 3.4. Yield

Maize grain yield (Yd) per plant (Figure 5) had differences between treatments (Tukey, *p* ≤ 0.05). The highest Yd (189.6 g) was obtained in the S2A2f1 treatment, similar to S2A1f1 and S2A0f1 (151.30 and 176.70 g, respectively), while the lowest Yd (27.5 g) was obtained in S1A1f0, which means that soil and fertilizer factors affected Yd (*F* = 519.48 and *F* = 290.80, respectively). The main effects graph for yield indicates that S2 and fertilizer application have similar effects on the maize grain yield. Yd was found to be associated (Table 5) with some yield components such as AP, DT and NP, with non-zero correlations and positive but weak linear trends (0.516, 0.394 and 0.392, respectively). This means that when AP, DT and NP increase, Yd tends to increase. Yd was also strongly correlated with AS (*r* = −0.820, *p* = 0.000), indicating that, as AS increases, Yd decreases. Ps correlated with Yd and most of the variables, indicating the indirect effect of inoculated AMF on phosphorus availability for maize plants.

## 4. Discussion

### 4.1. Substrate Characteristics

The soils used as substrates were different in nature. According to the Soil Survey Staff (SSS) [41] at the subgroup level, Typic Dystrusterts are soils with high clay contents, which are found in regions with dry temperate climates, are non-saline and have pH values lower than 4.5 [50]. Typic Ustifluvents, in contrast, are soils with minimal pedogenetic alterations, formed from sedimentation processes (resulting from the transport of fluvial currents) and are found in dry temperate climate zones [41,51]. The surface layer of both soils is considered non-saline [34]. However, the pH of these soils is different, such that, according to Porta et al. [52], the Typic Dystrustert under study is strongly acidic and the Typic Ustifluvent is neutral. Moreover, the discrepancy in AMF species composition in the native communities in each of the soils (three in Typic Dystrustert and nine in the Typic Ustifluvents) can be related to the distinguishing characteristics of the soils, such as the pH and clay content. In this regard, different authors have indicated that pH is one of the abiotic factors that affect the growth and development of AMF [53,54]; furthermore, pH can be conditioned by the content and type of clay [8,55]. It is important to highlight that the acidity or neutrality condition, the mineralogy of clay-sized particles and the taxonomic classification of soils are indicated at the subgroup or family level [41]. However, the existence of microorganisms such as AMF is not mentioned, information that could be useful for making soil management decisions (application of agrochemicals, organic amendments or biological ameliorators). For example, the presence of *Rhizophagus aggregatus*, *Funneliformis geosporum*, *Paraglomus occultum*, *Diversispora aurantia*, *Diversispora trimurales*, *Gigaspora candida*, *Gigaspora gigantea*, *Acaulospora mellea* and *Septoglomus* sp. in *Typic Ustifluvent* reflects the suitability of this soil to host these AMF species; this could be related to the pH (neutral), the CO and the available phosphorus content [56,57]. In contrast, the existence of *Funneliformis geosporum*, *Paraglomus occultum* and *Diversispora aurantia* in *Typic Dystrustert* shows the ability of these three species to establish symbiosis with maize plants under abiotic stress, given the pH condition and clay content of this soil, since the genera *Funneliformis*, *Paraglomus* and *Diversispora* have been reported to be tolerant to an acidic pH and high clay content [58,59,60]. The nine species found in the soils under study have been reported in nine vegetation types in Mexico [61], in particular, xerophytic shrublands, as well as in different agroecosystems and in other parts of the world [60,62]. This is consistent with what was found in the study, as the sites from which the soils (substrates) were obtained correspond to cultivated soils (agroecosystem) associated with mesquite and scrub vegetation [38] in the uncultivated areas.

### 4.2. Colonization and Sporulation

Colonization by AMF and their sporulation depend on the edaphic characteristics [18], the type of host plant, its stage of development and life cycle [63,64], crop management [4,57,65], the application of a single species or a consortium of these fungi [66] and the synergy or antagonism that may be established between AMF and other microorganisms native to these soils [67]. In this research, some of these situations were detected; the colonization and sporulation of *C. claroideum* and the consortium were influenced by the soil taxa (Typic Dystrustert and Typic Ustifluvent), the fertilized or non-fertilized condition of those soils and the presence of the native AMF, while the maize plant and its management remained constant. This suggests that the colonization and sporulation responses of allochthonous AMF in an agroecosystem will depend mainly on the physical, chemical and biological characteristics of each soil subgroup or family. In relation to the biological characteristics, the richness and quantity of the native AMF that can be found in different types of soil [12] become important since, depending on the edaphic environment in which the AMF are found, the effectiveness of the inoculated AMF would probably be enhanced. In this regard, Bender et al. [67] indicated that native AMF communities could create conditions conducive to the establishment of inoculated AMF. Thus, when native AMF communities are less abundant, the adaptation success of inoculated AMF is higher, which would be reflected in colonization and sporulation. The above could explain why the Typic Dystrustert with three native AMF species had, both with *C. claroideum* and with the consortium, a colonization on maize roots (>96%) higher than 70. 66% in the same soil subgroup, but uninoculated. In contrast, Typic Ustifluvent (with nine native AMF species and uninoculated) had 96.6% colonization, higher than when this soil was inoculated with *C. claroideum* and the AMF consortium (94.00–90.67%, respectively). This suggests that, in soil with numerous native AMF species, the probability of adaptation for inoculated AMF would decrease, presumably due to antagonistic and competitive interactions between native AMF and inoculated, allochthonous AMF [68]. If we consider the above, we can deduce that different soil subgroups could present different numbers of native AMF species and even that exogenous AMF strains applied in the soils would produce different colonization and sporulation for each species depending on the subgroup or family of a soil. This leads to the premise that knowledge of the soil at the subgroup or family level, together with information on its native AMF, would be useful in predicting the results of soil inoculation with exogenous AMF, or even whether it is necessary to do so. However, soil classification, even at the hierarchical family level, does not go into this detail.

### 4.3. Aggregates Stability

The AS of soil can vary depending on various biotic and abiotic factors [69], for example, the physical and chemical characteristics of the soil [70] and its native or added AMF [4,33]. According to Qin et al. [71], the AS increases when AMF are applied. In this regard, Liang et al. [72] indicated that mycorrhizal roots and mycorrhizal hyphae contribute mechanically to the aggregates stability by retaining them in the intricate networks they form and through the formation and excretion of glomalin [73]. Thus, AMF consistently contributes to the formation and stability of soil macroaggregates [74]. This behavior was observed in this study’s results since a tendency to increase stability was detected when soils were inoculated with exogenous AMF; however, this response was different in both soils since the AS differed. *Claroideoglomus claroideum* contributed to the increase in AS, mainly in the Typic Distrustert. It has been established that the genus *Claroideoglomus*, as well as other AMF, can improve soil structure and increase the stability of aggregates, mainly of the macroaggregate type (>0.25 mm) [75,76]. This is the opposite of what happened with the AMF consortium, which was less effective than that of the individual AMF. In this regard, Koziol and Bever [77] reported that the application of AMF mixtures improved soil conditions; however, the effectiveness of AMF can vary due to competition with native AMF in the soil [21], the addition of fertilizers [55] and the composition of species [78], as occurred in our study. This allows us to deduce that the stability of soil aggregates depends on the type of soil and native AMF, the type of AMF inoculum (a single species or a consortium) with which the soil is inoculated and even the management (fertilization or not) of the soil, as well as the relation established between native and allochthonous AMF. In this regard, Leifheit et al. [79] have indicated that the role of these fungi in the formation and stability of soil aggregates responds to multiple factors.

### 4.4. Yield

The presence of native or exogenous (inoculated) AMF in the soil leads to an increased maize yield [80,81]. This yield is related to the availability of nitrogen and phosphorus in the soil, caused by the symbiosis established by AMF with maize [82,83]. In this regard, Mena-Echevarría et al. [84] reported that inoculation with AMF (either as a single species or a consortium) efficiently promotes plant development and increases maize yield; for example, maize grain yield has been related to some yield components like plant height, with heights of 196.0 to 205.0 cm reported when the soil was inoculated with AMF [85]. This is in agreement with the results of our research since the maximum yields and plant heights were obtained both with the application of the consortium and when *C. claroideum* was applied in the Typic Ustifluvents, as well as with the native AMF community that was identified in such soil. The difference in yield between soils and the action of allochthonous AMF on them is due to the physical and chemical characteristics of soils [16] and the influence of native AMF [80]. The evidence from our study suggests that there is a relation between edaphic characteristics and species composition of native AMF communities, which naturally influence the activity of allochthonous AMF and the availability of phosphorus and nitrogen in the soil, which in turn influence maize growth and grain yield. However, the critical levels (below or above) of soil phosphorus and nitrogen contents at which AMF decrease their activity, or even cease it [86], and the relationships established between the different AMF species that make up the communities, depending on the physical, chemical and biological characteristics of the soil, are not clear.

## 5. Conclusions

The results of this research showed that two contrasting soils, which were used as substrates for the maize plants, presented communities of native arbuscular mycorrhizal fungi (AMF), integrated by a composition and richness of different species. Soil classified as fine-loamy, mixed, superactive, nonacid, thermic Typic Ustifluvent presented nine species of native AMF (*Rhizophagus aggregatus*, *Funneliformis geosporum*, *Paraglomus occultum*, *Diversispora aurantia*, *Diversispora trimurales*, *Gigaspora candida*, *Gigaspora gigantean*, *Acaulospora mellea* and *Septoglomus sp*), while the fine, mixed, semiactive, thermic Typic Dystrustert soil had three species (*Funneliformis geosporum*, *Paraglomus occultum* and *Diversispora aurantia*) of the nine that were found in the other soil. The native AMF in each soil influenced the activity of the allochthonous AMF added to the soil. The stability of the macro-aggregate-size aggregates (>0.25 mm) of each soil increased when they did not receive fertilization, since the addition of fertilizers decreases the activity of AMF (native or allochthonous) and, consequently, the stability of the aggregates. However, more research is needed to determine if this behavior is established in all soils and to determine the positive or negative interactions that occur between native and allochthonous AMF considering the physical, chemical and biological characteristics of the soils. Finally, the authors wish to highlight the importance of including the formative component, related to the presence of microorganisms in the soil, as is the case of AMF, at the family level in the “Keys to Soil Taxonomy” soil classification system, since it would lead to better and more efficient decision making on soil management.

## Figures and Tables

**Figure 1 microorganisms-11-02615-f001:**
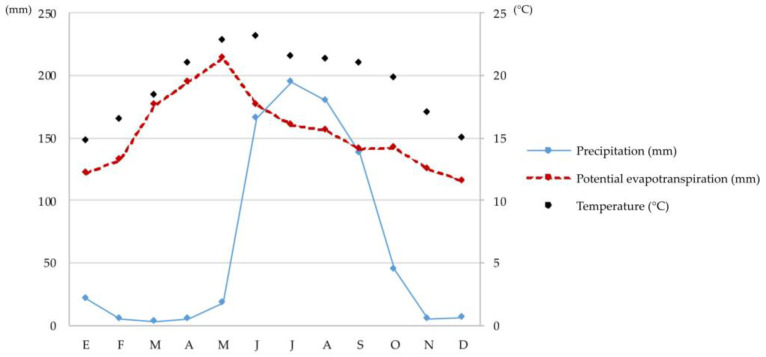
Climogram of Tlajomulco, Jalisco.

**Figure 2 microorganisms-11-02615-f002:**
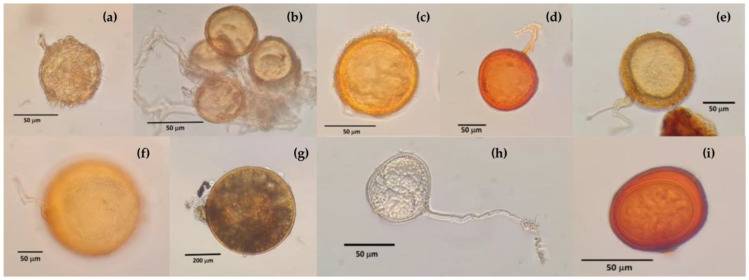
Spores of arbuscular mycorrhizal fungi identified in the soils used as a substrate: (**a**) Diversispora trimurales, (**b**) Rhizophagus aggregatus, (**c**) Diversispora aurantia, (**d**) *Funneliformis geosporum*, (**e**) *Septoglomus* sp., (**f**) *Gigaspora candida*, (**g**) *Gigaspora gigantea*, (**h**) *Paraglomus occultum* and (**i**) *Acaulospora mellea*.

**Figure 3 microorganisms-11-02615-f003:**
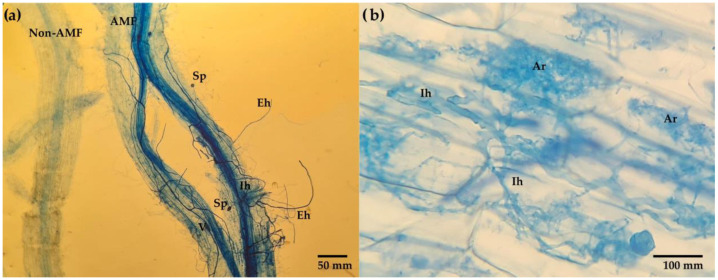
Mycorrhizal structures of corn roots at 16 weeks after planting. (**a**) Corn root colonized by AMF and (**b**) intradical structures. V vesicle, Sp spore, Ar arbuscule, Ih intercellular hyphae and Eh external hyphae.

**Figure 4 microorganisms-11-02615-f004:**
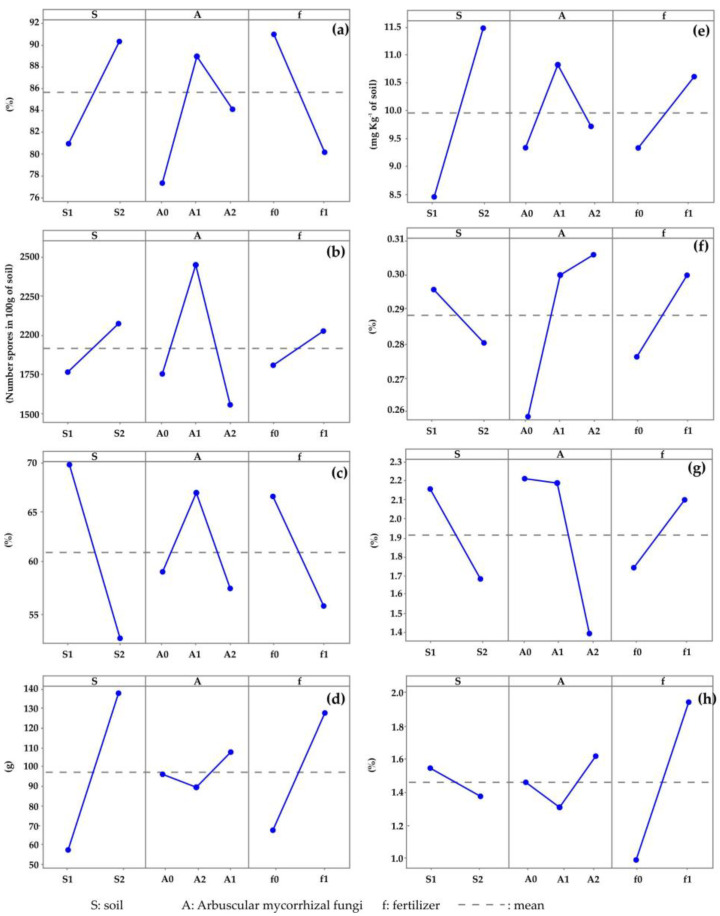
Graphs of main effects of the levels of each factor under study for (**a**) colonization, (**b**) sporulation, (**c**) aggregates stability, (**d**) yield of maize, (**e**) phosphorous in soil, (**f**) total nitrogen in soil, (**g**) phosphorous in plant and (**h**) total nitrogen in plant.

**Figure 5 microorganisms-11-02615-f005:**
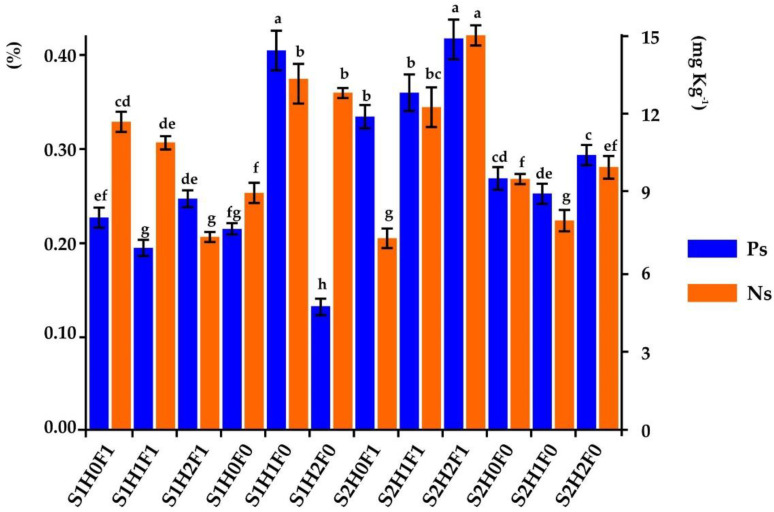
Content of available phosphorus and total nitrogen in the soils of the treatments (different letters in bars of the same color indicate differences between treatments (Tukey, *p* ≤ 0.05)).

**Table 1 microorganisms-11-02615-t001:** Physical and chemical characteristics of the soils that were used as substrates.

Soil	Hs^1^	Deep (cm)	OC (%)	pH	EC dS m^−1^	S (%)	L (%)	R (%)	Bd (g cm^−3^)	CEC (cmol_(+)_ Kg^−1^)	PBS (%)	P_2_O_5_ (mg Kg^−1^)
S1	Ap	0–20	1.09	4.40	0.89	3.26	46.98	49.76	1.34	13.45	34.80	93.04
	C1	20–45	1.03	4.35	0.52	1.80	48.12	50.08	1.36	11.22	19.32	52.93
	C2	45–72	0.59	6.31	0.47	1.10	59.86	39.04	1.53	12.10	38.83	6.42
	C3	>72	0.92	7.63	0.60	3.54	38.24	58.22	1.47	17.46	21.35	3.21
S2	Ap	0–16	1.64	6.87	0.98	44.60	32.04	23.36	1.34	15.41	57.12	140.40
	C1	16–45	0.83	5.62	0.16	40.28	36.36	23.36	1.32	16.51	60.30	116.15
	C2	45–65	0.46	6.28	0.33	59.72	23.32	16.96	1.15	9.10	30.41	16.40
	C3	65–82	0.75	6.80	0.34	65.52	17.48	20.00	1.11	11.40	41.73	6.12
	C4	82–95	0.48	6.61	0.38	58.64	21.68	19.48	1.29	11.20	40.79	12.43
	C5	>95	0.35	6.80	0.25	40.84	32.76	26.40	1.36	15.12	55.80	10.68

Hs^1^: horizon; OC: organic carbon; EC: electric conductivity, S: sand; L: lime; R: clay; Bd: bulk density; CEC: cationic exchange capacity; PSB: percent bases saturation.

**Table 2 microorganisms-11-02615-t002:** Species of native AMF in the soils that were used as substrates.

AMF	Typic Dystrustert	Typic Ustifluvent
*Rhizophagus aggregatus* (N.C. Schenck and G.S. Sm.) C. Walker		X
*Funneliformis geosporum* (T.H. Nicolson and Gerd.) C. Walker and A. Schluessler	X	X
*Paraglomus occultum* (C. Walker) J.B. Morton and D. Redecker	X	X
*Diversispora aurantia* (Błaszk, Blanke, Renker and Buscot) C. Walker and A. Schüßler	X	X
*Diversispora trimurales* (Koske and Halvorson) C. Walker and A. Schüßler		X
*Gigaspora candida* Bhattacharjee, Mukerji, J.P. Tewari and Skoropad		X
*Gigaspora gigantea* (T.H. Nicolson and Gerd.) Gerd. and Trappe		X
*Acaulospora mellea* Spain and N.C. Schenck		X
*Septoglomus* sp.		X

**Table 3 microorganisms-11-02615-t003:** Results of the variables that were evaluated in the treatments in studies.

Treat ^1^	N	Sp	Co	AS	Yd	PH	SD	Pp	Np
S1A0f0	5	866.00 h ^2^	70.66 cd	63.29 de	56.90 b	2.40 abcd	3.10 a	3.03 a	1.06 d
S1A1f0	5	2000.00 d	97.33 a	79.76 a	27.50 b	2.16 cd	2.78 b	3.00 a	1.08 d
S1A2f0	5	1570.00 f	96.67 a	72.58 b	36.00 b	2.16 cd	3.38 a	0.76 d	1.06 d
S2A0f0	5	2284.00 c	96.67 a	62.93 de	68.20 b	2.34 bcd	1.92 c	1.44 bcd	1.00 d
S2A1f0	5	3153.00 a	90.67 ab	67.86 bcd	76.09 b	1.95 d	1.92 c	1.50 abcd	0.60 e
S2A2f0	5	1760.00 e	94.00 ab	50.35 f	74.60 b	2.07 d	1.84 c	0.91 cd	1.13 d
S1A0f1	5	1492.00 f	59.32 d	62.27 e	55.40 b	2.51 abcd	2.78 b	2.55 ab	2.08 b
S1A1f1	5	2742.00 b	83.33 abc	68.52 bc	73.50 b	2.78 abc	3.10 ab	2.16 abcd	1.73 c
S1A2f1	5	1926.00 d	77.99 bc	64.89 cde	61.50 b	2.83 ab	2.68 b	1.42 bcd	2.43 a
S2A0f1	5	2342.00 c	82.67 abc	46.21 f	176.70 a	3.05 a	1.76 c	1.66 abcd	1.82 c
S2A1f1	5	1914.00 d	84.64 abc	50.51 f	151.30 a	3.08 a	1.82 c	2.28 abcd	1.94 bc
S2A2f1	5	1760.00 e	93.33 ab	40.45 g	186.90 a	3.02 a	1.88 c	2.32 abc	1.94 bc

^1^ Treat: treatments; N: number of replicas; Sp: spores’ density (number of spores in 100 g of soil); Co: colonization; AS: aggregates stability (%); Yd: yield (g); PH: plant height (m); SD: stem diameter (cm); Pp: phosphorus in plant tissue (mg kg^−1^); NP: total nitrogen in plant (%). ^2^ Different letters in the same column indicate significant differences between treatments (Tukey, *p* ≤ 0.05).

**Table 4 microorganisms-11-02615-t004:** Factorial analysis.

Factor	Soil		AMF		Fertilizer		Interactions			
Variable	*F* ^2^	*p*	*F*	*p*	*F*	*p*	S-A	S-f	f-A	S-A-f
Sp ^1^	674.00	0.000	2111.49	0.000	344.62	0.000	*			
Co	23.38	0.000	18.15	0.000	30.48	0.000				*
AS	573.96	0.000	68.95	0.000	243.23	0.000	*			
Ps	190,442.69	0.000	15,917.47	0.000	33,829.45	0.000		*		
Ns	4.10	0.048	15.19	0.000	9.13	0.004				*
Pp	6.74	0.012	9.76	0.000	3.90	0.054	*			
Np	50.17	0.000	59.44	0.000	1719.79	0.000			*	*
Yd	519.48	0.000	9.04	0.000	290.80	0.000		*		

Sp ^1^: spores’ density (number of spores in 100 g of soil); Co: colonization; AS: aggregates stability (%); Yd: yield (g); PH: plant height (m); SD: stem diameter (cm); Pp: phosphorus in plant tissue (mg kg^−1^); NP: total nitrogen in plant (%); *: significant interaction. ^2^ *F* = value of *F* test. *p* = probability of reject α ≤ 0.05.

**Table 5 microorganisms-11-02615-t005:** Correlation between the results of the study variables.

Variable		Co ^1^	Sp	AS	AP	DT	Ps	Ns	Pp	Np
Sp	*r* *p*	0.237 0.069								
AS	*r* *p*	0.015 0.075	0.537 0.050							
AP	*r* *p*	−0.215 0.099	0.061 0.644	−0.482 0.000						
DT	*r* *p*	−0.270 0.037	−0.222 0.644	0.654 0.000	−0.242 0.062					
Ps	*r* *p*	0.253 0.050	0.048 0.714	−0.485 0.000	0.290 0.024	−0.594 0.000				
Ns	*r* *p*	0.154 0.241	−0.212 0.105	−0.075 0.570	0.138 0.293	0.084 0.525	0.339 0.009			
Pp	*r* *p*	−0.232 0.075	−0.107 0.414	0.023 0.861	0.239 0.065	0.148 0.260	0.301 0.019	0.201 0.123		
Np	*r* *p*	−0.448 0.000	−0.086 0.515	−0.371 0.003	0.657 0.000	−0.008 0.954	0.170 0.193	0.148 0.258	0.131 0.317	
Yd	*r* *p*	0.101 0.444	0.433 0.049	−0.820 0.051	0.516 0.000	0.394 0.000	0.515 0.000	0.159 0.224	−0.141 0.282	0.392 0.002

^1^ Sp: spores’ density (number of spores in 100 g of soil); Co: colonization; AS: aggregates stability (%); Yd: yield (g); PH: plant height (m); SD: stem diameter (cm); Pp: phosphorus in plant tissue (mg kg^−1^); NP: total nitrogen in plant (%); *r*: correlation coefficient; *p*: value *p* = 0.05.

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
