# Peer review of "Arbuscular Mycorrhizal Fungi Associated with Maize (Zea mays L.) in the Formation and Stability of Aggregates in Two Types of Soil"

_microorganisms, 2023, doi:10.3390/microorganisms11112615_

Round 1
Reviewer 1 Report
Gómez-Leyva et al. describes a very interesting study, the authors did a lot of work, and the methodology used is adequate for the objectives of the study. The results are of interest and support the conclusions. That being said, the manuscript has the potential to be accepted. However, there is still some minor issues need to be addressed before the paper could be accepted as follows:
Comments
Lines 11-14: Please focus on the novelty of this work.
Lines 19-22: The presentation of the key findings of experimental results should be improved and data regarding the mainly measured indicators should be presented.
The manuscript is full of abbreviations, I suggest to include a list of abbreviations after the keywords.
Line 83: Give the coordinates.
Figure 1: What are P, PET, and T ???
Line 156: Give more details about the pots such as (height, diameter, etc…)
Line 161: Give the variety of maize plant.
Line 166: What is Da ???
Table 3: I suggest condensing the number of alphabetical of the statistics (For example: abcd to a-d)
Line 273: Italicize P throughout the manuscript
Line 306: What is SSS ???
Line 437: Do not cite a reference in the conclusion section
Kind Regards.
Minor editing of English language required
Author Response
RESPONSES TO COMMENTS AND OBSERVATIONS TO THE MANUSCRIPT
MICROORGANISMS-2672364
REVIEWER 1: The authors thank you for your patience, reading and analysis of the article, and above all for sharing your experience and knowledge, since your comments, suggestions and doubts helped to improve the writing.
Lines 11-14: Please focus on the novelty of this work.
- Thanks for the comment. The summary was written again, taking into account his comment.
Lines 19-22: The presentation of the key findings of experimental results should be improved and data regarding the mainly measured indicators should be presented.
- Thanks for the comment. The summary was written again, taking into account his comment.
The manuscript is full of abbreviations, I suggest to include a list of abbreviations after the keywords.
- Thanks for your suggestion. However, in the body of the writing each of the abbreviations that were used are defined.
Line 83: Give the coordinates.
- Thanks for the comment. The coordinates of the municipality were noted in the corresponding paragraph and appear in red.
Figure 1: What are P, PET, and T ???
- Thanks for your question. The definition of P, PTE and T were defined as part of the figure.
Line 156: Give more details about the pots such as (height, diameter, etc…)
- Thanks for your comment. The details about height and diameter of the pots were noted in the e corresponding paragraph and appear in red.
Line 161: Give the variety of maize plant.
- Thanks for your comment. The maize plant used was a hybrid that was mentioned in line 133.
Line 166: What is Da ???
- Thanks for your comment. Sorry, Da should be Bd (Bulk density) as it was defined. The Bd appears in red.
Table 3: I suggest condensing the number of alphabetical of the statistics (For example: abcd to a-d)
- Thanks for your question. Thank you for your comment. The proposal is interesting, because in this way more space could be used between the columns of the table, where the letters that correspond to groupings (separation of means) of the means would appear. However, I am not sure that it is correct, which is why the authors have decided that this part remains as originally proposed.
Line 273: Italicize
- Thanks for your comment. P throughout the manuscript, were italicized and appears in red.
Line 306: What is SSS ???
- Thanks for your comment. SSS sss means Soil Survey Staff. The name Soil Survey Staff appears in the corresponding paragraph and is abbreviated to SSS, and appear in the corresponding paragraph in red.
Line 437: Do not cite a reference in the conclusion section
- Thanks for your comment. We delete the appointment you indicated.

Reviewer 2 Report
This article presents a very broad experimental review on the population of different species of arbuscular mycorrhizal fungi that can present beneficial associations in two types of soil used for pots where a species of corn can be grown.
Within the context of the work, it can be mentioned that the topic is interesting because it provides more information about the factors that affect the populations of mycorrhizal fungi, as well as the different environmental conditions that affect them to achieve optimized development.
I found it interesting that the search for information was done through an experimental design, a methodology that is very useful when it is required to verify different experimental parameters and their effect within an analytical process.
The introduction presents sufficient preliminary information to understand the foundation and purpose of the scientific work developed by the authors. Among the references used, some self-citations by the authors were found, but they were not considered excessive, given the total number of references used. Likewise, an evaluation was made of probable plagiarism, but nothing was found that indicated it, so it is considered that this is a completely original work.
It is worth mentioning that the experiments within the experimental design were adequately directed, the information presented on the experimental methodologies carried out is perfectly founded both in its proposal within the experimental determinations, as well as in the particular application.
The images presented are perfectly visible and discriminative from each other, allowing that the descriptions made by the authors to be adequately understood. It is worth mentioning that the quality of the images themselves is excellent. The graphs perfectly show the results obtained.
The statistical tools used show consistent results within the analysis carried out by the authors.
It is considered that the article has the necessary qualities to be published without major modifications.
Author Response
RESPONSES TO COMMENTS AND OBSERVATIONS TO THE MANUSCRIPT
MICROORGANISMS-2672364
REVIEWER 2: Thank you for all your comments, all of them allow us to continue striving to improve our work. Thanks again and best regards.
